# The Contraindications to Combined Oral Contraceptives among Reproductive-Aged Women in an Obstetrics and Gynaecology Clinic: A Single-Centre Cross-Sectional Study

**DOI:** 10.3390/ijerph19031567

**Published:** 2022-01-29

**Authors:** Ghadah A. Assiri, Douha F. Bannan, Ghadah H. Alshehri, Manal Alshyhani, Walaa Almatri, Mansour A. Mahmoud

**Affiliations:** 1Department of Clinical Pharmacy, College of Pharmacy, King Saud University, P.O. Box 2454, Riyadh 11451, Saudi Arabia; 436925577@student.ksu.edu.sa (M.A.); 436200613@student.ksu.edu.sa (W.A.); 2Pharmacy Practice Department, King Abdulaziz University, Jeddah 80260, Saudi Arabia; dbannan@kau.edu.sa; 3Pharmacy Practice Department, College of Pharmacy, Princess Nourah Bint Abdulrahman University, Riyadh 84428, Saudi Arabia; ghALShehri@pnu.edu.sa; 4Clinical and Hospital Pharmacy Department, College of Pharmacy, Tibah University, Al-Madinah Al-Munawarah 42353, Saudi Arabia; mammm.99@gmail.com

**Keywords:** contraindication, drug, contraceptive, oral, combined, obstetrics-gynaecology department, hospital, estimated proportion, risk factors

## Abstract

This study aimed to determine the estimated proportion of contraindications among women taking combined oral contraceptives (COCs) and to assess the risk factors associated with their contraindications. This study was cross-sectional. Reproductive-aged women (18–49 years) on any COCs between 2018 and 2020 were recruited from one obstetrics–gynaecology clinic in a university-affiliated hospital and were included. Contraindications were defined using the World Health Organization (WHO) Medical Eligibility Criteria (MEC) for Contraceptive Use. Data were collected from electronic medical records for all included women, as well as a standardised, pretested, structured survey for one-third of the women. In this cross-sectional study, 380 women using COCs were included. Their mean age was 31.645 ± 7.366 years. Among them, 131 (34.5%) participated via a survey and electronic records, while the other 249 (65.5%) participated via electronic records only. The majority of the participants had a Bachelor’s degree (59.0%) and were married (62.1%). The overall estimated proportion of patients with at least one contraindication to COCs according to category 3 (relative contraindications) or 4 (absolute contraindications) was 31.3% (95% CI 26.63–35.99). The most common contraindications observed were controlled hypertension, category 3 (12.1%); major surgery with prolonged immobilisation, category 4 (4.7%); migraine with aura at any age, category 4 (4.2%); breastfeeding from six weeks to less than six months postpartum, category 3 (4.0%); and diabetes mellitus with complications, category 4 (3.2%). Significant factors associated with contraindications to COCs were married women (OR 2.19, 95% CI 1.38–3.46), those aged 35 years or more (OR 2.33, 95% CI 1.49–3.66), and those with one or more live births (OR 2.19, 95% CI 1.38–3.46). Ensuring proper assessment prior to prescribing and considering alternatives suitable for long-term use among women taking an oral contraceptive regularly is recommended.

## 1. Introduction

Combined oral contraceptives (COCs), also known as birth control pills, are a hormonal therapy that contain both oestrogen and progestin. They are used mainly for contraception, as well as for several non-contraceptive indications such as polycystic ovarian syndrome (PCOS) [1] and dysmenorrhea [2]. Prevention of ovulation is considered the dominant mechanism of action [3].

Despite the benefits of COCs, due to the increasing risk of adverse effects in the profile of oestrogen and progestin combination therapy, there are some medical conditions considered as contraindications. Contraindications are a common prescribing error [4] and are defined as “a circumstance, condition, symptom, or factor that increases the risk associated with a medical procedure, drug, or treatment” [5]. Therefore, medical conditions should be carefully considered when selecting appropriate candidates for this therapeutic strategy.

In Saudi Arabia, women can only obtain COCs through prescription, whereas in some other countries (e.g., China and India) [6], COCs are available over the counter, which adds a greater risk of harm to the patient. A study conducted in a community setting in Jazan, Saudi Arabia, investigated the utilisation pattern and safety profile of oral contraceptives, and reported that out of 496 women, two-thirds (69.0%) were using or had used COCs. In addition, more than half of the participants (69.6%) experienced at least one side effect [7]. In another retrospective study, 39 out of 108 women with cerebral venous thrombosis (36.1%) were secondary to oral contraceptive pills [8].

Most of the studies conducted in Saudi Arabia have focused on attitude, use, and knowledge of oral contraceptives [9,10,11,12,13]. Although many studies have examined the proportion or rate of contraindications to COCs globally [14], some studies have only investigated one contraindication [15,16], and another study investigated four contraindications [17]. However, there are still no studies in Saudi Arabia. Contraindications are considered to be a serious health problem, based on the World Health Organization (WHO) Medical Eligibility Criteria (MEC) for Contraceptive Use [18], and they must be considered before one designs interventions to prevent contraindication-related adverse events. Therefore, we were interested in assessing the estimated proportion of contraindications to COCs among reproductive-aged women in an obstetrics–gynaecology (OB-GYN) clinic in a university-affiliated hospital in central Saudi Arabia, and assessing the risk factors associated with contraindications to COCs. The uniqueness of this study is that it investigated most of the contraindications, and also used two methods for data collection to ensure the comprehensiveness and accuracy of the data.

## 2. Materials and Methods

### 2.1. Reporting

This study manuscript followed the Strengthening the Reporting of Observational studies in Epidemiology (STROBE) checklist and the reporting of studies conducted using observational routinely-collected health data (RECORD) statement [19,20] (Appendix A).

### 2.2. Ethical Considerations

Ethical approval was obtained from the Institutional Review Board (IRB) at King Saud University (KSU) (Research project no.: E-20-5243) (Appendix A). Every patient record had a code number for use in the research. In addition, an agreement was made with each woman participating in the study. To ensure anonymity and confidentiality, only patient identifiers (name, file number, phone number, etc.) were used for the research. Permission to use the WHO MEC for Contraceptives Use was obtained.

### 2.3. Study Design

A retrospective cross-sectional study was undertaken to measure the estimated proportion of contraindications to COCs among reproductive-aged women in an OB-GYN clinic.

### 2.4. Setting

The data were collected from an OB-GYN clinic in a university-affiliated hospital in the central region of Saudi Arabia. The university-affiliated hospital is a 1200-bed multi-disciplinary facility with general and subspecialty medical services that provides primary, secondary, and tertiary care to its patients. In addition, it provides a designated outpatient and inpatient facility, state-of-the-art surgical services, a fully equipped and staffed laboratory, radiology, pharmacy services, a dedicated home healthcare programme, and other support services [21].

### 2.5. Participants

Reproductive-aged women (18–49 years) on COCs with or without a known contraindication to the use of COCs were included in this study. In addition, women who used COCs for contraceptive or non-contraceptive health benefits such as to reduce heavy menstrual bleeding (HMB), menstrual pain, improve acne, endometriosis, hirsutism and menstrual irregularities, PCOS, and premenstrual syndrome (PMS) were included [22]. Women who were on other hormonal therapies besides COCs and postmenopausal women were excluded.

### 2.6. Variables

#### 2.6.1. Baseline Characteristics

The baseline variables included were age, body mass index (BMI), educational level, marital status, number of births, and type of COCs. Two types of COCs were available in the hospital formulary: drospirenone and oestradiol hemihydrate (Angeliq^®^) and drospirenone and ethinylestradiol (Yasmin^®^). Other COCs were obtained from outside the hospital, such as cyproterone acetate and ethinylestradiol (Diane^®^), drospirenone and ethinylestradiol (Midian^®^), and desogestrel and ethinylestradiol (Regulon^®^, Marvelon^®^). These COCs were checked against the Saudi Food and Drug Authority (SFDA) list of human medications and were subsequently classified into prescription medications [23].

#### 2.6.2. Contraindication Outcomes

The WHO MEC contains four categories for contraceptive eligibility. Category 1 is a condition for which there is no restriction for the use of the contraceptive method [18]. Category 2 is a condition where the advantages of using the method generally outweigh the theoretical or proven risks [18]. Category 3 (relative contraindication) is a condition where the theoretical or proven risks usually outweigh the advantages of using the method [18]. Category 4 (absolute contraindication) is a condition that represents an unacceptable health risk if the contraceptive method is used [18]. Here, the outcomes were based on categories 3 and 4 of the WHO MEC for Contraceptives Use (fifth edition), which represent contraindications [18].

The relative contraindications are the following: postpartum for less than 21 days without other risk factors for venous thromboembolism (VTE) or postpartum between 21 and 42 days with other risk factors for VTE, breastfeeding between six weeks and six months postpartum (primarily breastfeeding), age 35 or older and smokes less than 15 cigarettes/day, history of hypertension where blood pressure cannot be evaluated (including hypertension in pregnancy), adequately controlled hypertension where blood pressure can be evaluated, systolic blood pressure of 140–159 mmHg or diastolic blood pressure of 90–99 mmHg, migraine headaches without aura at age ≥35 years, past and no evidence of current breast cancer for five years, past or current gall bladder disease, past COC-related cholestasis, anticonvulsant therapy (phenytoin, carbamazepine, barbiturates, primidone, topiramate, oxcarbazepine, or lamotrigine), or antimicrobial (rifampicin or rifabutin) [18].

The absolute contraindications are the following: postpartum for less than 21 days with other risk factors for VTE, breastfeeding from less than 6 weeks postpartum, age 35 or older and smokes ≥15 cigarettes/day, vascular disease, hypertension systolic blood pressure ≥160 mmHg or diastolic blood pressure ≥100 mmHg, history or acute deep vein thrombosis/pulmonary embolism (DVT/PE), DVT/PE and established on anticoagulant therapy, major surgery with prolonged immobilisation, migraine headaches with aura (at any age), current breast cancer, severe (decompensated) cirrhosis, malignant liver tumours, hepatocellular adenoma, known thrombogenic mutation, current and history of ischemic heart disease, stroke, complicated valvular heart disease (pulmonary hypertension, risk of atrial fibrillation, and history of subacute bacterial endocarditis), or systemic lupus erythematosus (SLE) with positive (or unknown) antiphospholipid antibodies [18].

The categories that should be assessed according to the severity of the condition are the following: (a) diabetes with nephropathy/retinopathy/neuropathy or with another vascular disease, diabetes of a >20-year duration and (b) acute liver disease or flare viral hepatitis [18].

#### 2.6.3. Risk Factors

The risk factors associated with COC contraindications are age, BMI, educational level, marital status, and parity [17,24]. Parity refers to the number of pregnancies exceeding 20 weeks’ gestation and provides information regarding the outcome of each pregnancy [3].

### 2.7. Data Collection/Data Source

All electronic medical records for those women who visited the OB-GYN clinics between 1 January 2018 and 15 December 2020 were requested. After removing duplicated records, each record was given a code number and the inclusion/exclusion criteria were applied until the required sample size was reached.

Data were extracted from both electronic medical records and from a standardized, pretested, structured survey.

The data collected from the electronic medical records were age, educational level, marital status, past medical history, medication history, and height/weight, which were needed to calculate BMI. BMI is a measure of total body weight relative to height, and can provide a better assessment of total body fat than weight alone. Those with a BMI of 18.5–24.9 kg/m^2^ were considered to have “normal” weight. The terms overweight, obese, and severely obese were reserved for those with a BMI of 25–29.9 kg/m^2^, 30–39.9 kg/m^2^, and ≥40 kg/m^2^, respectively [25].

The survey was developed by four pharmacists, considering its clarity and relevance to ensure content validity. The face validity was tested on a pilot of five women. The survey was conducted by two trained pharmacy research interns (M.A. and W.A.), under the supervision of the principal investigators (G.A.A. and M.A.M.), either by a phone call or online link via phone messages according to the most convenient method for the women after obtaining their agreement to participate in the study.

The survey consisted of three sections: (a) the first section consisted of demographic information, (b) the second consisted of the COC type, and finally, (c) the third included a checklist to determine the presence/absence of category 3 or 4 contraindications to COCs according to the WHO MEC [18]. The survey language was English, and an experienced translator back-translated it to Arabic [26] (Appendix A).

The survey was used to ensure that the all of the outcome variables of interest were available for the research, in case they were not available from the medical record or the record was not updated. For example, the date of the last delivery, breastfeeding status, smoking history, and pattern of migraine headaches.

### 2.8. Bias

Non-response bias is common in survey research [27]. To overcome this problem, several approaches were taken, such as making the survey short, clear, and easy to respond to and through follow-up calls to non-respondents. In addition, a self-reporting survey was used to collect data. This may result in recall bias. As a result, we also collected data from electronic records to further reduce recall bias.

### 2.9. Sample Size Estimation

The sample size was based on the assumption that the proportion of reproductive-aged women with contraindications to combined oral contraceptives is 50%, due to the fact there were no previous similar studies from Saudi Arabia [28]. The minimum sample size required for the current study was calculated using the following formula [28]:*N* = Z ^2^ × P (1 − P)/d^2^
where:

*N* = sample size;

Z = Z statistic (for a level of confidence of 95%, Z = 1.96);

P = expected prevalence (in proportion to 1, P = 0.50);

d = precision (in proportion to 1, d = 0.05);

*N* = 1.96^2^ × 0.50 (1 − 0.50)/0.05^2^.

*N* = 380 reproductive-aged women.

### 2.10. Statistical Methods

The continuous variables are presented as the mean ± standard deviation (SD), and the categorical variables as number and percentage. The association between the demographic variables such as age, BMI, educational level, marital status, and parity and the dependent variable (contraindications to COCs) was assessed using univariate logistic regression. The statistical analysis was conducted using STATA Software SE/14.0 (StataCorp LLC, Station, TX, USA).

### 2.11. Data Access and Cleaning Methods

The Excel datasheet was checked for errors in data, outliers, and missing data.

## 3. Results

### 3.1. Demographic Characteristics

Table 1 represents the demographic characteristics. Out of the 704 records for the women who visited the OB-GYN clinics in the study period, 380 records met the inclusion criteria and were retrieved. The mean age of the 380 women was 31.645 ± 7.366 years. Data were collected for 131 (34.5%) women via electronic records and surveys, while for the remaining 249 (65.5%) women, data were collected via electronic records only. The majority of the participants had a Bachelor’s degree (59.0%) and were married (62.1%). Regarding the types of COCs, 88.7% of the women used drospirenone and ethinylestradiol, 7.9% used desogestrel and ethinylestradiol, and 3.4% used cyproterone acetate and ethinylestradiol.

### 3.2. The Estimated Proportion of Contraindications to COCs

Table 2 represents the estimated proportion of contraindications to COCs. The overall estimated proportion of women with at least one contraindication to COCs according to categories 3 and 4 of the WHO MEC was 119/380 (31.3%; 95% CI 26.63–35.99). The remaining 261 women had no contraindications.

Out of the 119 women with contraindications, the majority (*n* = 90) had one contraindication (23.7%), 17 had two contraindications (4.5%), 7 had three contraindications (1.8%), 3 had four contraindications (0.8%), 1 had five contraindications (0.3%), and 1 had six contraindications (0.3%).

The estimated proportion of women with at least one contraindication to COCs according to category 3 of the WHO MEC (relative contraindication) was 85/380 (22.4%; 95% CI 18.16–26.58). The estimated proportion of women with at least one contraindication to COCs according to category 4 of the WHO MEC (absolute contraindication) was 57/380 (15.0%; 95% CI 11.39–18.61).

The most common contraindications observed for category 3 were controlled hypertension (12.1%; 95% CI 8.81–15.40) and breastfeeding from six weeks to less than six months postpartum (4.0%; 95% CI 1.98–5.91). The most common contraindications observed for category 4 were major surgery with prolonged immobilisation (4.7%; 95% CI 2.59–6.88), migraine with aura at any age (4.2%; 95% CI 2.18–6.24), and diabetes mellitus with complications (3.2%; 95%CI 1.39–4.92).

### 3.3. Risk Factors Associated with Contraindications to COCs

Table 3 represents the risk factors associated with the contraindications to COCs using univariate logistic regression. Based on the estimated odds ratio (OR), women aged 35 years or more or those with one or more live births were at a significantly increased risk (by two times) of contraindications to COCs (OR 2.33, 95% CI 1.49–3.66, *p* < 0.001 and OR 2.19, 95% CI 1.38–3.46, *p* = 0.001, respectively). Married women were at an 82% greater risk of contraindications to COCs (OR 1.82, 95% CI 1.14–2.91, *p* = 0.012). There were no statistically significant associations between women with contraindication to COCs with regards to BMI and educational level.

## 4. Discussion

In this research, we aimed to determine the estimated proportion of contraindications to COCs among reproductive-aged women, as well as the risk factors associated with their contraindications. The overall estimated proportion of contraindications to COCs according to categories 3 and 4 of the WHO MEC was 119/380 (31.3%; 95% CI 26.63–35.99). The risk factors associated with these contraindications were married women aged 35 years or more or women with one or more live births.

Relative contraindications were found in 22.4% of women. In these cases, the risks of using COCs outweigh the benefits, especially if an alternative is not available or is unacceptable for the patient; thus, COCs can be prescribed with caution and follow-up to ensure the safety of its use. Absolute contraindications were found in 15.0% of the women, which represents an unacceptable health risk if the contraceptive method is used.

The most common contraindications observed for category 3 were controlled hypertension (12.1%; 95% CI 8.81–15.40) and breastfeeding from six weeks to less than six months postpartum (4.0%; 95% CI 1.98–5.91). For category 4, they were major surgery with prolonged immobilisation (4.7%; 95% CI 2.59–6.88), migraine with aura at any age (4.2%; 95% CI 2.18–6.24), and diabetes mellitus with complications (3.2%; 95% CI 1.39–4.92).

One of the most common contraindications observed for category 3 was controlled hypertension (12.1%). This was expected, as the prevalence of hypertension in Saudi Arabia is high [29]. According to the May Measurement Month (MMM) 2019 global screening initiative of the International Society of Hypertension, of a total of 25,023 males and females, 29.2% had hypertension [29].

Evidence has shown that the risk of cerebral thromboembolism increases significantly in women with diabetes mellitus, hypertension, migraine, and past thromboembolic events [30]. If oral contraception is the choice, women with these increased thrombotic risks should use oestrogen-containing oral contraceptives after careful considerations of the risks, if there are any [30].

Knowledge of contraindications to COCs among young females in one city in Saudi Arabia was assessed by Alsulaiman et al. (2017). A total of 40.9% (174/426) of the women mentioned that they should not use COCs if they had severe hypertension, 42.5% mentioned that they must wait at least six weeks after giving birth before starting COCs, 38.0% said that COCs can cause clotting in the legs and lungs, and 50.9% were aware that they must stop using COCs if they developed migraines with aura [31]. However, the mean scores regarding knowledge of contraindications were low (score: 2), indicating a poor level of knowledge among young females in Riyadh (a score of 1 represents very poor knowledge and 5 represents excellent knowledge) [31].

A recent systematic review looked at the available evidence on the prevalence of combined hormonal contraceptive (CHC) use among women with contraindications according to the WHO recommendations (2015) [14]. The systematic review showed that the prevalence ranged from 5.9% to 41.9% [14]. Although the prevalence in our study was within the systematic review’s overall prevalence range, it was not feasible to draw a comparison with the systematic review’s overall prevalence because (a) the studies in the review used oral, patch and vaginal ring contraceptive methods compared to oral contraceptives only in our study, and (b) the outcomes or numerators/denominators were inconsistent between the different studies included in the review and our study. One consistent finding was found in the review in comparison to our study, which was that the most frequent contraindication was systemic hypertension.

In order to make comparisons with other studies possible, the studies need to have the same numerators and denominators as our study. In this study, the estimated proportion of contraindications was analysed as the prevalence of contraindications among women who used contraceptives. We were able to compare our results with two studies. The first study was conducted in Brazil to estimate the prevalence of contraindicated use of oral contraceptives using a survey [17]. The study found that the prevalence of contraindications was 11.7% (95% CI 10.6–13.7) [17]. This study’s proportion is lower than our results; however, the prevalence in said study may have been underestimated because only systemic hypertension, smokers and ≥35 years, diabetes mellitus, and cardiovascular disease were investigated, and the type of oral contraceptive used was not specified. Two consistent findings were found in the aforementioned study in comparison to our study, which are that the most frequent contraindication was systemic hypertension (9.1%; 95% CI 7.6–10.8) and its prevalence was higher in women aged 45–49 years when compared to younger women (adjusted prevalence ratio = 7.30; 95% CI 4.2–12.8) [17].

The second study was a prospective cohort study in North America. Women were asked about their desired methods of contraception and medical history (self-reported medical contraindications). The proportion was lower than our results, as only 24 of 1010 women desiring CHC (2.4%; 95% CI 1.53–3.52%) were found to have true medical contraindications to CHCs using a chart review [32]. However, only women who desired a CHC (they were not actually using the contraceptive) were included, and this might have resulted in an underestimation of their prevalence.

Our study has several strengths. First, in this study, we investigated most of the contraindications to provide a better estimate of their prevalence. Second, we used the most updated WHO MEC for Contraceptive Use 2015 [18] to inform and categorise the contraindications. Third, we used two data collection methods, namely, a survey and medical records, to ensure the comprehensiveness and accuracy of the data.

The limitations of this study include, first, that the study was conducted in one centre and the women included had mostly attended high school or had Bachelor’s degrees, which might be not representative of all women in Saudi Arabia and this makes generalising the results difficult. Second, we relied on self-reported blood pressure or one reading from an electronic medical record instead of taking the average of two blood pressure measurements. Third, there were other contraindications that we could not categorise according to the severity, such as liver disease, and this could have resulted in an underestimation in our findings, although we suspect that this contraindication is rare. Finally, this study did not examine the adverse effects on the involved women; therefore, we do not know what harmful effect, if any, that the contraindications had on these women.

When prescribing contraceptives to women, physicians have a central role in screening COCs for contraindications, explaining the risks and potential side effects, weighing the risks versus benefits, as well as providing advice on the appropriate use of contraceptives.

The physician who prescribes the COC or the pharmacist who dispenses could provide all women with a formal written risk assessment for contraindications. In addition, we should not ignore the role of self-care; women can self-screen using a checklist of contraindications [33]. Promoting the importance of proper assessment prior to prescribing and considering the alternatives suitable for long-term use such as the copper intrauterine device or progestin-only depot medroxyprogesterone acetate or progestogen-only pill, since they have fewer contraindications than the combined pill among women regularly taking a contraceptive, is recommended [18].

More research could be conducted on the accuracy of contraindications by using different data collection methods, such as surveys, medical records, or self-screening, and more research could be carried out in different regions in Saudi Arabia to increase generalisability.

## 5. Conclusions

The estimated proportion of contraindications to COCs was 31.3%. The significant factors associated with contraindications to COCs were married women aged 35 years or more or with one or more live births. Weighing the risks versus benefits and screening COCs for contraindications in order to ensure the safe use of COCs is recommended.

## Figures and Tables

**Table 1 ijerph-19-01567-t001:** Demographic characteristics.

	Categories	Total Sample Size, *N* = 380
Age	-	Mean 31.645 ± 7.366
	<35 years	252 (66.3%)
	≥35 years	128 (33.7%)
Data source	Electronic medical records only	249 (65.5%)
	Electronic medical records and survey	131 (34.5%)
Education	Unschooled	3 (0.8%)
	Elementary school	4 (1.1%)
	Secondary school	3 (0.8%)
	High school	139 (36.6%)
	Diploma	5 (1.3%)
	Bachelor’s degrees	224 (59.0%)
	Postgraduate	2 (0.5%)
Body mass index (kg/m^2^)	Normal: 18.5–24.9	124 (32.6%)
	Overweight: 25–29.9	132 (34.7%)
	Obese: 30–39.9	111 (29.2%)
	Severely obese: ≥40	13 (3.4%)
Marital status	Single	144 (37.9%)
	Married	236 (62.1%)
Number of births	0 births	163 (42.9%)
	≥1 births	217 (57.1%)
Type of combined oral contraceptive	Drospirenone and ethinylestradiol	337 (88.7%)
	Desogestrel and ethinylestradiol	30 (7.9%)
	Cyproterone acetate and ethinylestradiol	13 (3.4%)

Abbreviations: N, number; SD, standard deviation; COCs, combined oral contraceptives.

**Table 2 ijerph-19-01567-t002:** Estimated proportion of each and overall category 3 or 4 contraindications to combined oral contraceptives *.

Condition	Category 3 MEC (Relative Contraindications) *n* (%; 95% CI)		Category 4 MEC (Absolute Contraindications) *n* (%; 95% CI)	
Postpartum	<21 days: Without other risk factors for venous thromboembolism (VTE)	8 (2.1%; 0.66–3.56)	<21 days: With other risk factors for VTE	1 (0.3%; −0.25 to 0.78)
	Between 21 and <42 days: With other risk factors for VTE	9 (2.4%; 0.83–3.90)		
Breastfeeding	6 weeks to <6 months postpartum (primarily breastfeeding)	15 (4.0%; 1.98–5.91)	<6 weeks postpartum	8 (2.1%; 0.66–3.56)
Smoking	Age 35 or older and smokes <15 cigarettes/day	2 (0.5%; −0.20 to 1.26)	Age 35 or older and smokes ≥15 cigarettes/day	0
Hypertension	History of hypertension, where blood pressure cannot be evaluated (including hypertension in pregnancy) Adequately controlled hypertension, where blood pressure can be evaluated Systolic 140–159 mmHg or diastolic 90–99 mmHg	46 (12.1%; 8.81–15.40)	Systolic ≥160 mmHg or diastolic ≥100 mmHg	1 (0.3%; −0.25 to 0.78)
Deep vein thrombosis/pulmonary embolism (DVT/PE)	-		History of DVT/PE or acute DVT/PE	8 (2.1%; 0.66–3.56)
			DVT/PE and established on anticoagulant therapy	5 (1.3%; 0.16–2.47)
			Major surgery with prolonged immobilisation	18 (4.7%; 2.59–6.88)
Headaches	Migraine without aura at age ≥35 years	2 (0.5%; −0.20 to 1.26)	Migraine with aura at any age	16 (4.2%; 2.18–6.24)
Diabetes	-		Nephropathy/retinopathy/neuropathy Other vascular disease or diabetes of a >20-year duration **	12 (3.2%; 1.39–4.92)
Others	Past or current gall bladder disease	5 (1.3%; 0.16–2.47)	Known thrombogenic mutation	4 (1.1%; 0.02–2.08)
	Anticonvulsant therapy: Certain anticonvulsants (phenytoin, carbamazepine, barbiturates, primidone, topiramate, and oxcarbazepine) Lamotrigine	2 (0.5%; −0.20 to 1.26)	Current and history of ischemic heart disease	3 (0.8%; −0.10 to 1.68)
	Antimicrobial therapy: Rifampicin or rifabutin therapy	1 (0.3%; −0.25 to 0.78)	Systemic lupus erythematosus (SLE) with positive (or unknown) antiphospholipid antibodies	3 (0.8%; −0.10 to 1.68)
Overall	Women with at least one contraindication according to category 3			85 (22.4%; 18.16–26.58)
	Women with at least one contraindication according to category 4			57 (15.0%; 11.39–18.61)
	Women with at least one contraindication according to category 3 or 4			119 (31.3%; 26.63–35.99)

Abbreviations: COC, combined oral contraceptive; DVT, deep vein thrombosis; MEC, Medical Eligibility Criteria; PE, pulmonary embolism; SLE, systemic lupus erythematosus; VTE, venous thromboembolism. * Adopted from the World Health Organisation (WHO) Medical Eligibility Criteria (MEC) for Contraceptive Use (18). ** The category should be assessed according to the severity of the condition. Contraindications not reported in this table were not identified in this study.

**Table 3 ijerph-19-01567-t003:** Factors associated with contraindications to combined oral contraceptives (data obtained from univariate logistic regression).

Factor	Category	OR	*p*-Value (95% CI)
Age	Less than 35 years old	2.33	0.000 * (1.49–3.66)
	35 years or more		
Body mass index	Overweight (25.0–29.9 kg/m^2^)	0.815775	0.466 (0.47–1.41)
	Obese (30.0–39.9 kg/m^2^)	1.544171	0.115 (0.89–2.65)
	Severely obese (>40 kg/m^2^)	1.045045	0.944 (0.30–3.61)
Educational level	Diploma or high school or below	1.145833	0.912 (0.10–12.93)
	University degree or above	0.7730061	0.835 (0.07–8.68)
Marital status	Single	1.82	0.012 * (1.14–2.91)
	Married		
Parity	0 live births	2.19	0.001 * (1.38–3.46)
	1 or more live births		

Abbreviations: OR, odd ratio; CI, confidence interval. * *p*-value significant if <0.05.

## Data Availability

All available data, such as the study protocol, raw data, or programming code, can be obtained by contacting the corresponding author.

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
