# Peer review of "The Contraindications to Combined Oral Contraceptives among Reproductive-Aged Women in an Obstetrics and Gynaecology Clinic: A Single-Centre Cross-Sectional Study"

_ijerph, 2022, doi:10.3390/ijerph19031567_

Round 1

Reviewer 1 Report

Generally, the study is of average quality and interest.

Moreover, the text needs English editing. 

In the Methods, the authors calculate a minimum sample size of 380 women based on the assumption that the proportion of reproductive-aged women with contraindications to combined oral contraceptive pills is 50%, which is extremely high. I consider that this calculation could be omitted. After all, the sample is not representative, since it is from a single center and included women are mostly overweight or obese and with university education. 

In the results, the authors use precisions of two or three decimal digits in the percentages. Since the total number of subjects in the study is 380, every subject counts for 100/380=0.26%. This means that there can be no differences less than 0.26% among any proportions. Thus, the use of only one decimal digit is enough. 

In the end of the results, the three risk factors were derived from a univariate regression analysis. 

Overall, I suggest acceptance after minor revision, however I consider it of moderate priority for the Journal.

Reviewer 2 Report

Contraindicatons as well as side effects are comonly known and I can't see any particular purpose to publish this data. Manuscript is well written clear and elegant, but may be it should be addressed to phisicians in Saudi Arabia and be published in local medical journal

Author Response

Thank you.

Reviewer 3 Report

The paper by Assiri et al. reported a cross-sectional study concerning the proportion of contraindications among women who are taking COCs and to assess risk factors associated with their contraindications between 2018-2020 in a single center. Three significant factors associated with contraindications to COCs were highlighted as 1) married women 2) Age >= 35 years 3) has >= 1 live birth. The authors suggested that it is important to prior assess the factors and contraindications before prescribing long-term regular use of COCs.

This manuscript seems to be clear, structured, and has adequate ethics statements. However, there are some limitations of this study should be considered:

  1. The recall bias – a weakness of cross-sectional study may need to be thoroughly considered and discussed in this study, apart from the ‘Non-response bias’ mentioned in the Materials and Methods.
  2. The novelty and significance of this study would be suggested to highlight in the manuscript.
  3. If the three significant factors associated with contraindications to COCs should be assessed, options of alternative forms of contraception would be considered and discussed.

Reviewer 4 Report

I do not understand the relevance of the conclusion: the associated risk factors cannot be the status of a married woman and one or more children born alive. These may come from statistical calculations, but must be judged clinically. The good conclusion would be the advice that contraceptives should be prescribed by a doctor all over the world, because the study proves that a third of women take contraceptives, although they have contraindications that can aggravate their health.

Round 2

Reviewer 1 Report

In the revised manuscript that I see, the authors did not edit the percentages to one decimal digit.